# aPEAch: Automated Pipeline for End-to-End Analysis of Epigenomic and Transcriptomic Data

**DOI:** 10.3390/biology13070492

**Published:** 2024-07-02

**Authors:** Panagiotis Xiropotamos, Foteini Papageorgiou, Haris Manousaki, Charalampos Sinnis, Charalabos Antonatos, Yiannis Vasilopoulos, Georgios K. Georgakilas

**Affiliations:** 1Laboratory of Genetics, Section of Genetics, Cell Biology and Development, Department of Biology, University of Patras, 26504 Patras, Greece; 2Information Management Systems Institute, ATHENA Research Center, 15125 Marousi, Greece

**Keywords:** next-generation sequencing, NGS, RNA-seq, miRNA-seq, ATAC-seq, ChIP-seq, differential gene expression, peak calling, gene-set enrichment, clustering

## Abstract

**Simple Summary:**

The emergence of next-generation sequencing (NGS) signified a revolution in biology research by capturing the significance of DNA loci and RNA molecules on a genome-wide scale. The complexity and volume of NGS data highlight the need for robust and user-friendly computational tools. The framework presented here, aPEAch, is an automated pipeline for end-to-end analysis of DNA and RNA sequencing assays, including small RNA sequencing. Implemented in Python, it allows users to customize the analysis and results, handling single or multiple replicates in batches, while also automating advanced unsupervised learning analyses. aPEAch offers quality control reports, fragment size distribution plots, and intermediate files supporting reproducibility and interoperability, along with publication-ready visualizations.

**Abstract:**

With the advent of next-generation sequencing (NGS), experimental techniques that capture the biological significance of DNA loci or RNA molecules have emerged as fundamental tools for studying the epigenome and transcriptional regulation on a genome-wide scale. The volume of the generated data and the underlying complexity regarding their analysis highlight the need for robust and easy-to-use computational analytic methods that can streamline the process and provide valuable biological insights. Our solution, aPEAch, is an automated pipeline that facilitates the end-to-end analysis of both DNA- and RNA-sequencing assays, including small RNA sequencing, from assessing the quality of the input sample files to answering meaningful biological questions by exploiting the rich information embedded in biological data. Our method is implemented in Python, based on a modular approach that enables users to choose the path and extent of the analysis and the representations of the results. The pipeline can process samples with single or multiple replicates in batches, allowing the ease of use and reproducibility of the analysis across all samples. aPEAch provides a variety of sample metrics such as quality control reports, fragment size distribution plots, and all intermediate output files, enabling the pipeline to be re-executed with different parameters or algorithms, along with the publication-ready visualization of the results. Furthermore, aPEAch seamlessly incorporates advanced unsupervised learning analyses by automating clustering optimization and visualization, thus providing invaluable insight into the underlying biological mechanisms.

## 1. Introduction

Next-generation Sequencing (NGS) technologies have created a transformative era in the field of genomic research and medical applications [1]. Characterized by continuous advancements, they have swiftly become the Swiss army knife for researchers in all biological disciplines. Technologies for digitizing DNA and RNA sequences have inspired researchers to develop protocols for the isolation and study of genomic loci that participate in biological processes and transcriptome profiles that establish the function, phenotype, and thus the identity of cells. The continuous advancements in the field have led to a significant decrease in costs per sequenced megabase of DNA, enabling research groups to adopt sequencing experiments to an even greater extent than ever before [2].

While comprehensive analysis frameworks such as GALAXY [3] and the ENCODE project set of pipelines [4] offer valuable tools for genomic research, practical considerations and user experience limitations exist. GALAXY stands out for its widespread adoption and user-friendly features, providing a web-based interface that facilitates the execution of bioinformatic workflows without requiring extensive programming expertise. The platform’s support for workflow management allows users to construct analysis paths, incorporating established bioinformatic tools. However, the web-based nature introduces challenges, including a potential learning curve for seasoned users accustomed to command-line interfaces. Moreover, the necessity to upload data to the web, poses difficulties in handling large and intricate datasets, impacting the platform’s efficiency for certain analyses. Although a local host option exists, the setup and operation of a GALAXY server can be challenging, even for experienced users. These practical considerations highlight the need for researchers to carefully evaluate the suitability of GALAXY based on the specific requirements and complexities of their genomic analyses.

ENCODE provides an extensive guideline for the analysis of several genomic assays, which are implemented within its own framework of standardized pipelines that contribute to the reproducibility principles [4]. ENCODE pipelines are well known for guiding users toward a standardized and reproducible outcome analysis path that provides meaningful insights into functional genomic elements accumulated from a plethora of sequencing protocols. Although ENCODE provides gold standard guidelines concerning the analysis of various protocols, their implementation lacks user flexibility in the analysis, and it is relatively difficult even for experienced users.

A variety of tools and suites have been developed over the years for NGS data analysis for specific experimental protocols. Cloud-based applications, such as CSI NGS Portal [5] and Taverna2 [6], offer more user-friendly approaches for inexperienced users. However, scalability, data security, and privacy are some disadvantages to consider. Regarding software that can be deployed locally, there are solutions that cover parts of NGS data analysis requirements, such as NARWHAL [7], which can handle the preprocessing and mapping part, and metaseqR2 [8], which aggregates different algorithms for the differential gene expression analysis part. Furthermore, there exist end-to-end analysis implementations, such as DNAscan2 [9] and MethylStar [10]. However, those tools support specific NGS protocols, and they lack the analytic capacity of a broader range of experiments.

We introduce aPEAch, a modular in silico framework for analyzing the NGS data derived from a variety of protocols including ATAC-, ChIP-, and RNA-seq assays (Figure 1). Sequencing experiments typically produce samples with DNA or RNA fragments that need to be quality-assessed and mapped to the reference genome prior to any downstream analysis that extracts biological insights, such as differential gene expression analysis (DGEA) for RNA-seq, gene-set enrichment analysis, and clustering of genes and/or samples for the identification of functional groups. The modular architecture of aPEAch enables the user to create an analysis path specifically tailored to the properties of each experimental assay or even to apply multiple paths to explore alternative analytic approaches. aPEAch was designed to be easy to use and is accompanied by rich documentation and extensive tutorials that thoroughly describe the usage of each module. It is a framework that can support a wide range of analyses and is suitable for both experienced and novice users. aPEAch is open source for academic usage and freely accessible from GitLab (https://gitlab.com/a5465/apeach). The version used to perform the analysis presented in this manuscript corresponds to 28 June 2024.

## 2. Materials and Methods

The aPEAch architecture is composed of seven modules that can be used to form distinct analysis pipelines tailored to experimental assays with different requirements. Each module requires an initialization file as input that includes all the necessary information required for the execution of the module’s components, such as software parameters, and a flag that signifies whether a specific component will be executed or skipped. The first module also specifically requires the filesystem location of the FASTQ formatted files of each sample organized in a single folder, with distinguishable names as described in the documentation: the same prefix for files associated to the same sample (i.e., replicates and/or paired-end samples), distinct suffixes for paired-end (_R1/R2), and clearly labeled for different replicates (i.e., rep1/2). aPEAch represents a sequential application of software components in the form of a pipeline; thus, the results are provided in a well-defined directory structure, with distinct subdirectories for each module and software component. This approach facilitates effortless exploration of results and straightforward addition of modules or software components in the aPEAch stack. Users are advised to consult the well-documented resources for each external software supported by aPEAch prior to its application to experimental samples for the selection of the optimal parameters depending on the corresponding assays’ properties.

### 2.1. Preprocessing and Mapping for DNA- and RNA-Sequencing Assays (PAM)

The first aPEAch module is related to the initial and most important stage of any NGS analysis pipeline, also known as raw read preprocessing and mapping. This step is common to both RNA-/DNA-seq, and it encompasses a series of steps starting with the evaluation of sample quality, followed by contaminant removal, and trimming of low confidence bases. The remaining reads are subsequently aligned to the reference genome or transcriptome. Currently, aPEAch supports FastQC [11] (v.0.12.1) and Minion [12] (v.15.065) for assessing sample quality, STAR [13] (v.2.7.10) for read alignment, and Picard [14] (v.3.0.0) for the removal of duplicate reads. Lastly, users may choose to exclude reads aligned to the mitochondrion.

### 2.2. Peak Calling (PC)

For the characterization of reproducible peaks in each sample, MACS2 (v2.2.7.1.1) software [15] is used in the peak calling module. Users are required to specify their preferred parameters for MACS2 and, if available, provide a control file. The “callpeak” sub-command is then executed, utilizing as input the previously generated file from the preprocessing and alignment module. Filtering of the input file is carried out based on user-defined options outlined in the initialization file. Commonly used parameters include the p-value or q-value cutoffs, as well as defining upper and lower limits for model building.

### 2.3. Peak Annotation (PA)

To assign peaks to genes, the nearest gene approach is commonly utilized by annotation tools such as ChIPpeakAnno [16]. This method involves associating a peak with the gene whose transcription start site (TSS) is closest to the peak’s genomic coordinates. However, this approach can be misleading as binding sites may be situated between the start sites of different genes.

To address this limitation, a novel method was developed and incorporated into aPEAch for accurately identifying the genomic placement of peaks. Users are required to provide a GFF3 file containing organism-specific annotation information and the preferred promoter length as an additional parameter. By integrating the GFF3 annotation file and the NARROWPEAK file from the peak calling module, the precise characterization of each peak can be determined.

Initially, the method examines if the peak is located within a gene region. If it is, further analysis is conducted to determine whether the peak is situated in a promoter or in an exon region. The promoter region is defined as a contiguous DNA sequence extended both upstream and downstream of the TSS. If the peak is within a gene but neither in a promoter nor in an exon region, it is classified as an intronic peak. If the peak’s coordinates do not overlap with any annotated genomic coordinates in the provided GTF file, it is considered to be intergenic.

### 2.4. Counts on Genomic Elements (COGE)

The quantification of reads aligned to functional genomic loci (i.e., peaks or genes) is a crucial step in generating the count matrix that captures the expression profile of these elements across samples. This matrix serves as the input for subsequent modules such as the differential gene/peak expression analysis module and the clustering module. To achieve this, aPEAch employs HTSeq [17] (v.2.0.3) to tally the reads by parsing the alignment files and checking the overlaps with the gene annotation provided in GTF format, in the case of RNA-seq data. For DNA-related data, a two-step process is followed, where the NARROWPEAK files (as generated by the peak calling module) from all samples are merged and a unique peak identifier is assigned to each newly formed peak. Subsequently, a GTF file is generated for counting reads on the respective intervals using HTSeq, resulting in a count matrix that serves as input for differential loci (peak) expression analysis and clustering modules.

### 2.5. Differential Loci Expression Analysis (DEA)

When comparing samples derived from different conditions, the identification of dysregulated loci is a crucial step. To seamlessly integrate this analysis into aPEAch, we developed a dedicated module that utilizes the widely adopted DESeq2 (v.2.0.3) algorithm [18,19], which is a staple in such analyses. This module performs normalization of the count matrix and conducts a comprehensive differential gene expression analysis. As quality control metrics, we apply principal component analysis (PCA), volcano, and MA plots to enhance the understanding of the identified dysregulated loci as well as to provide deeper insight into the differential loci expression analysis.

### 2.6. Automated Clustering Analysis (ACA)

The clustering module was developed to elucidate the patterns underlying gene expression or peak localization within a given dataset. This methodology leverages the count matrix generated by the COGE module, with the primary objective of grouping those elements into clusters based on their expression profile similarity. The initial step involves pairwise distance calculation across data points representing genes or genomic intervals, serving as input for the clustering algorithms. K-means and agglomerative algorithms are utilized to distinguish groups based on normalized read counts or q-values (only applicable to DNA-related samples). The visualization of formed clusters is represented as a heatmap and in the two-dimensional space, leveraging the t-SNE transformation.

### 2.7. Gene-Set Enrichment Analysis (GSEA)

The process of extracting biological information from the data analysis relies on the GSEA module. This module takes a set of dysregulated genes, as determined by differential gene expression analysis or the presence of peaks on gene promoters, and uses them as input for the GSEA, which is based on GSEApy [20] (v.1.1.3). The module searches user-defined databases for gene ontology terms and biological pathways associated with the specified genes, calculates the enrichment significance, and identifies whether specific functional categories or pathways are disproportionately represented within the list of genes of interest.

### 2.8. Application of aPEAch in Different Use Cases as Proof of Concept

aPEAch was applied on publicly available data from the Immunological Genome Project (ImmGen) consortium [21] and ENCODE project [22] to showcase the robustness and versatility of the framework in three distinct use cases (Table 1). We utilized ATAC-seq and RNA-seq datasets from CD4+- and CD8+-positive thymocytes, as well as miRNA-seq data from three tissues (heart, lung, liver) of *Mus musculus* postnatal day zero, to investigate the factors differentiating these cell types across various biological signals. The high-quality datasets provided by the ImmGen consortium and ENCODE served as an excellent foundation for our exploration of chromatin accessibility, transcriptomic profiles, and microRNA expression patterns. Moreover, these datasets also served as an ideal testbed for our algorithm, effectively demonstrating its efficacy in diverse molecular profiling applications.

The mouse reference genome and the corresponding gene annotation were downloaded from Ensembl v109 [23,24].

## 3. Results

aPEAch was developed with a user-centric strategy to efficiently manage the results generated by the various modules. This systematic structure streamlines data storage and enhances user experience. The creation of a dedicated project folder tailored to the user’s project specifications serves as a central repository for all project-related data. This versatile approach enables the reuse of the project folder for multiple runs of protocols, providing a comprehensive overview of the project’s progression. Within the parent project directory, individual sample subdirectories ensure the clarity and separation of the results for projects involving multiple samples. To further enhance data management, the results are categorized into module-specific subdirectories within each sample directory. This hierarchical storage frame enhances precision and data retrieval. Above all, the aPEAch system was designed to be user-friendly, simplifying the process of accessing and storing results and reducing the risk of errors.

### 3.1. aPEAch Application on RNA-Seq Data

RNA-sequencing data obtained from CD4+ and CD8+ single-positive naïve thymocytes were utilized in this study to demonstrate the RNA-seq pipeline modules (Figure 2). The dataset consisted of two distinct cell types, with two replicates for each type. The initial step involved the quality assessment of the raw data, followed by the alignment of the reads to the mouse reference genome (mm10). The quantification of the gene expression levels was achieved with the COGE module. The resulting count matrix served as the template for the differential gene expression analysis module, which identified genes exhibiting statistically significant over- or under-expression between the two cell types. To achieve a deeper understanding of the biological relevance of the gene expression changes, the GSEA module revealed the biological pathways in which the deregulated genes were prominently involved.

### 3.2. Chromatin Accessibility Exploration with aPEAch

aPEAch was also utilized to analyze ImmGen-derived ATAC-Seq data on DN1, DP, CD4+ and CD8+ single-positive T cells, which describe the regulatory landscape during the T-cell maturation process. Raw reads were initially subjected to quality assessment prior to their alignment on the mouse reference genome (mm10). Subsequently, the peak calling module was employed to identify read enriched peaks throughout the genome, suggesting functional open chromatin loci. The COGE module was utilized to quantify the reads associated with each peak, facilitating further analysis by the clustering and peak annotation modules, leading to the identification of chromatin accessibility patterns that probably shape the regulatory landscape responsible for T-cell maturation.

### 3.3. miRNA-Seq Analysis Using aPEAch

A dataset from the ENCODE project was used to showcase the capabilities in miRNA sequencing analysis leveraging aPEAch’s modules. Mouse embryo liver, heart, and lung samples (day 0) served as the input to the PAM module. The reads were assessed for adapter contamination and quality trimming prior to being mapped against the hairpin sequences extracted from miRbase, allowing zero tolerance for mismatches. The reads that did not align to the hairpin sequences were aligned to the reference genome, allowing one mismatch to account for mutations or the presence of isomiRs. Both count matrices were combined and forwarded to the clustering module to exploit the expression patterns on different subsets of miRNAs (Figure 3). 

## 4. Discussion

aPEAch stands out as a versatile and modular framework specifically developed to facilitate the analysis of a broad spectrum of NGS assays, encompassing both single- and paired-end protocols. Its intuitive design reflects the minimal input required from users and is tailored precisely to each assay’s unique specifications, ensuring user-friendliness without compromising analytical depth. A critical aspect of aPEAch is related to its focus on reproducibility, achieved through standardized analysis pipelines stipulated within user-defined initialization files.

While aPEAch excels in accommodating diverse sequencing protocols, it currently lacks full optimization for single-cell assays, variant calling analyses, the identification of novel miRNAs, and meta-analyses that combine data from multipurpose omics assays for the extraction of biological information. Moreover, the framework’s modules presently offer support for only a limited selection of tools publicly available in the research domain. Despite these limitations, aPEAch remains a dynamic platform, continuously undergoing updates and enhancements that enrich the framework with analytic functionalities that go beyond the concept of typical pipelines in the field and that are capable of catering to the evolving needs of cutting-edge research.

Despite its current constraints, aPEAch is a computational framework with novel components, empowering users of all proficiency levels to effortlessly execute sophisticated analyses, such as miRNA-mediated gene expression regulation, genomic classification of identified peaks based on existing annotation, and the identification of functional gene groups with unsupervised approaches, within reasonable timeframes. Notably, the novel clustering module further highlights aPEAch’s utility by automating a task that traditionally posed challenges even for seasoned bioinformaticians, thereby enhancing its analytical capabilities and expanding its potential applications.

## 5. Conclusions

In the era of big data in biology, where datasets are increasingly complex and voluminous, tools like aPEAch offer researchers structured and efficient approaches for extracting meaningful insights from their data. By providing a modular and user-friendly interface, aPEAch enables researchers to navigate the complexity of NGS data analysis with ease, regardless of their expertise. This democratization of data analysis not only accelerates the pace of research but also fosters collaboration and knowledge exchange within the scientific community.

Moreover, aPEAch’s commitment to reproducibility ensures the reliability and integrity of research findings, which are crucial for building upon existing knowledge and driving scientific discovery forward. By bridging the gap between data generation and interpretation, software frameworks like aPEAch are indispensable tools for exploiting the potential of NGS technology and advancing our understanding of complex biological systems.

## Figures and Tables

**Figure 1 biology-13-00492-f001:**
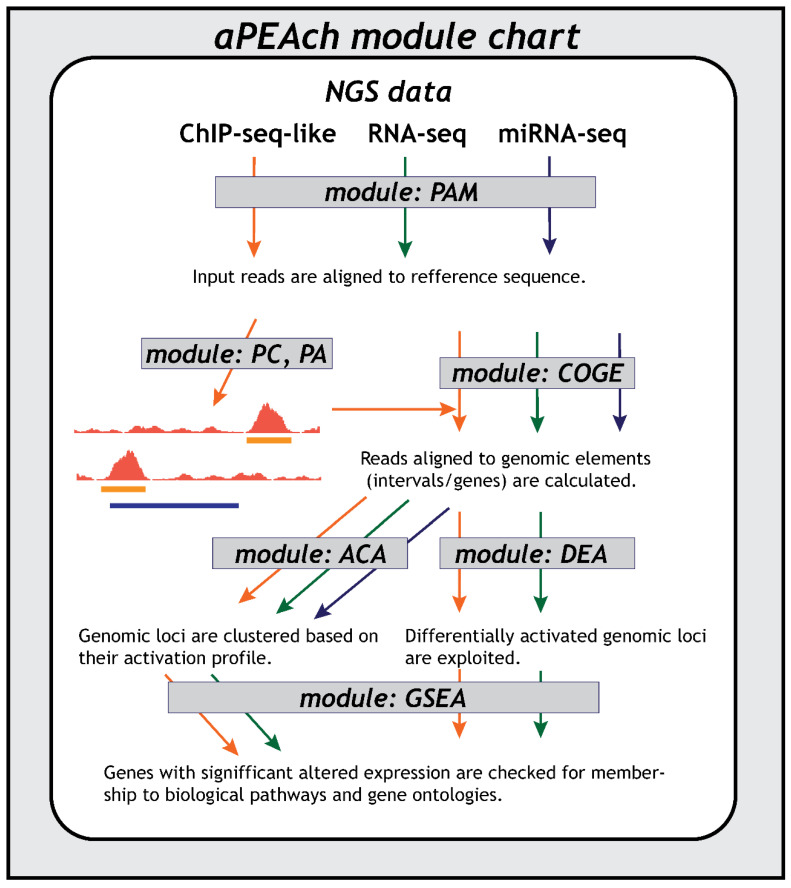
Overview of aPEAch’s module hierarchy and data flow depending on the NGS protocol; ChIP-seq-like (i.e., ATAC-, DNase-, MNase-seq) in orange, RNA-seq in green, miRNA-seq in blue).

**Figure 2 biology-13-00492-f002:**
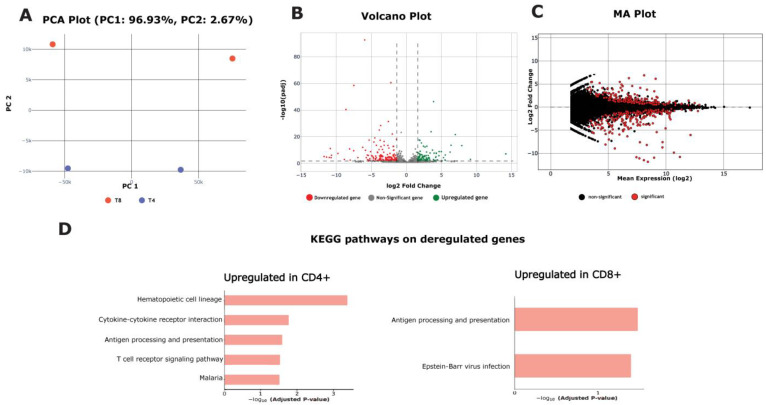
RNA-seq use case module’s results. (**A**) PCA plot including the processed samples (T4 cells in blue color and T8 cells in orange color), (**B**) volcano plot, in which dotted lines represent log2 fold change and adjusted *p*-value cutoffs, and (**C**) MA plot derived from the differential gene expression analysis. (**D**) KEGG pathways enriched with deregulated genes (upregulated in CD4+ T cells on the left, and upregulated in CD8+ T cells on the right).

**Figure 3 biology-13-00492-f003:**
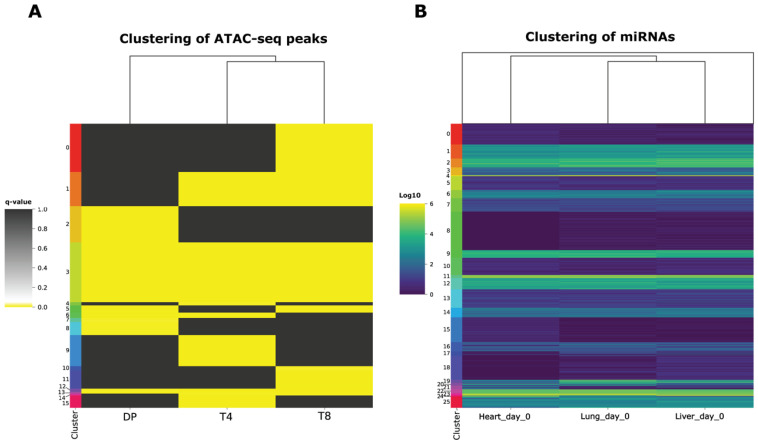
Clustering analysis results. (**A**) Clustering of ATAC-seq peaks based on the corrected p-value (log_10_ scale) as calculated by Macs2, for double-positive (DP), CD4 (T4), and CD8 (T8) single-positive T cells. (**B**) Clustering of miRNAs based on their expression (normalized counts based on DESeq2 variance stabilization normalization) profile across heart, lung, and liver samples on day 0. For both (**A**,**B**), the number of clusters was automatically calculated using the optimal value of the silhouette coefficient score.

**Table 1 biology-13-00492-t001:** Datasets used to showcase the application and efficacy of aPEAch.

Use Case	Sample	Accession Number
RNA-seq	T.4.Th#1_RNA-seq	GSM2932607
	T.4.Th#2_RNA-seq	GSM2932608
	T.8.Th#1_RNA-seq	GSM2932611
	T.8.Th#2_RNA-seq	GSM2932612
ATAC-seq	preT.DN1.Th#1	GSM2692171
	T.DP.Th#1	GSM2692334
	T.4.Th#1	GSM2692182
	T.8.Th#1	GSM2692184
miRNA-seq	lung_day_0	ENCSR161PUT
	liver_day_0	ENCSR456CZD
	heart_day_0	ENCSR246CIO

## Data Availability

Data are available in public repositories. The ENCODE project (www.encodeproject.org, accessed on 10 March 2024) for the miRNA-seq use case and ImmGen (www.immgen.org, accessed on 28 April 2023) for RNA- and ATAC-seq use cases. The code including aPEAch’s modules is available from GitLab (https://gitlab.com/a5465/apeach). The version used to perform the analysis presented in this manuscript corresponds to 28 June 2024.

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
