# Peer review of "aPEAch: Automated Pipeline for End-to-End Analysis of Epigenomic and Transcriptomic Data"

_biology, 2024, doi:10.3390/biology13070492_

Round 1

Reviewer 1 Report

Comments and Suggestions for Authors

This a novel study that will be very useful to the community and I would recommend publication. Some minor comments for author consideration are listed below.

1. ln 210: The ensembl version is not provided

2. I would encourage the authors to use pipeline management software like snakemake. Since snakemake is also in Python it should be a straightforward transition.

3. Figure 1: The fonts are too small and difficult to read

4. A README file would greatly help users to understand the use cases and installation process.

5. The prefix directive needs to be removed from the yaml file.

Reviewer 2 Report

Comments and Suggestions for Authors

Panagiotis Xiropotamos et al. introduce a pipeline named aPEAch, an automated pipeline for end-to-end analysis of epigenomic and transcriptomic data. Reading through the paper, my concerns are:

There is no detailed information about how to use the aPEAch tools in the paper. I suggest the authors publish the code and the user manuals on GitHub.

The aPEAch pipeline can only process bulk sequencing datasets; however, single-cell sequencing technology is widely used around the world now, so the novelty of aPEAch is not sufficient.

There are many other similar tools that can process NGS data as aPEAch does. The authors should clarify what the unique features of aPEAch are compared to other tools.

The authors showed the ability of aPEAch to process RNA-seq, miRNA-seq, and ATAC-seq in the main figures. However, the biggest challenge for users is not processing these datasets, but integrating the multi-omics data to gain biological insights. Adding some downstream analysis functions would largely improve the pipeline.

Reviewer 3 Report

Comments and Suggestions for Authors

Xiropotamos et al. presented a method, aPEAch, for automatic and integrative multi-omic data analysis. The method is of potential interest and usability due to the fast-growing volume of omics data. However, the authors didn’t provide a comprehensive demonstration of results from each module in aPEAch, and a detailed tutorial was lacking to install and implement the pipeline. Unfortunately, the paper cannot be considered for publication in its current form.

Major points to address:

1.        For each module in aPEAch, please clarify and provide demo figures to showcase the functions and expected results. For example, the authors can refer to a recent pipeline paper (PMID: 38091963).

2.        For the online tutorial, please provide detailed guidance to install and run the separate components in the method, e.g. provide codes and input data for each step. It’s also recommended to distribute aPEAch on Github, as it’s a much more mature platform.

Minor points to revise:

1.        Increase the resolution of all figures, use vector plot if possible.

2.        Line 230, please indicate clearly which Figure it refers to

Comments on the Quality of English Language

Minor editing of English language require

Round 2

Reviewer 2 Report

Comments and Suggestions for Authors

The author addressed all the questions.

Reviewer 3 Report

Comments and Suggestions for Authors

The authors have addressed my comments and made significant improvements to the manuscript. Thus, I recommend accepting the paper for publication.